# Substituent-Controllable Cascade Regioselective Annulation of *β*-Enaminones with *N*-Sulfonyl Triazoles for Modular Access to Imidazoles and Pyrroles

**DOI:** 10.3390/molecules28114416

**Published:** 2023-05-29

**Authors:** Hua Wang, Tongtong Zhou, Mengdi Wu, Qingqing Ye, Xinwei He

**Affiliations:** 1Key Laboratory of Functional Molecular Solids, Ministry of Education, Anhui Laboratory of Molecule-Based Materials (State Key Laboratory Cultivation Base), College of Chemistry and Materials Science, Anhui Normal University, Wuhu 241000, China; huaw2011@ahnu.edu.cn (H.W.); m18895330986@163.com (T.Z.); 2221011404@ahnu.edu.cn (M.W.); 2Department of Medicine, Chuzhou City Vocation College, Chuzhou 239000, China

**Keywords:** annulation, *β*-enaminones, imidazoles, pyrroles, *N*-sulfonyl trizaoles

## Abstract

A controllable synthesis of trisubstituted imidazoles and pyrroles has been developed through rhodium(II)-catalyzed regioselective annulation of *N*-sulfonyl-1,2,3-trizaoles with *β*-enaminones. The imidazole ring was formed through a 1,1-insertion of the N-H bond to α-imino rhodium carbene, followed by a subsequent intramolecular 1,4-conjugate addition. This occurred when the *α*-carbon atom of the amino group was bearing a methyl group. Additionally, the pyrrole ring was constructed by utilizing a phenyl substituent and undergoing intramolecular nucleophilic addition. The mild conditions, good tolerance towards functional groups, gram-scale synthesis capability, and ability to undergo valuable transformations of the products qualify this unique protocol as an efficient tool for the synthesis of *N*-heterocycles.

## 1. Introduction

Nitrogen-containing heterocycles are privileged structural motifs in various natural products and bioactive compounds [1,2]. Among them, imidazole and pyrrole frameworks are very common structural units widely distributed in natural products, pharmaceutics, agrochemicals, and other functional materials [3,4]. For this reason, the synthesis of such compounds continues to be a hot topic in modern synthetic chemistry [5,6,7,8]. Consequently, a large number of new reactions have been developed to construct structurally diverse imidazole and pyrrole derivatives, such as multicomponent reactions [9,10]. [3 + 2] cycloaddition [11,12,13,14], as well as both metal-catalyzed intermolecular [15,16] and intramolecular [17,18] cyclization strategies. Despite all the achievements, the development of efficient methods for their synthesis, particularly regiocontrolled synthesis of those containing multiple substituents from readily accessible compounds, is of ever-increasing importance.

In the past decades, 1,2,3-triazoles have emerged as capable precursors for the synthesis of various nitrogen heterocycles [19,20]. Upon treatment with rhodium(II) catalysts, *N*-sulfonyl-1,2,3-triazoles readily undergo denitrogenation reactions to form α-imino rhodium carbenes, a versatile intermediate that could promote a wide range of transformations [21,22]. In addition to common reactivities such as cyclopropanation [23,24], X-H insertion [25,26,27,28], and ylide formation [29,30,31], α-imino rhodium carbenes can also serve as [1C]- or aza-[3C]-synthons in stepwise cycloadditions, leading to the formation of various N-heterocycles [32,33,34,35]. As a major part of our research efforts in developing new methodologies for the construction of heterocycles [36,37,38,39], we herein describe an efficient strategy for the regioselective synthesis of trisubstituted imidazoles and pyrroles. This strategy involves the cascade N-H insertion to α-imino rhodium carbene, followed by substituent-controllable intramolecular annulation (Figure 1). In this scheme, the α-imino rhodium carbene acted as [2C] and aza-[3C]-synthons, respectively.

## 2. Results and Discussion

The optimization of a one-pot procedure for the formation of imidazole 3a from triazole **1a** and *β*-enaminone **2a** was undertaken (Table 1). Screening of various transition-metal catalysts revealed that dirhodium catalysts Rh_2_(OAc)_4_ and Rh_2_(oct)_4_ were demonstrated to be more efficient than other metal catalysts for this reaction (Table 1, entries 1–6). Further investigation showed that a lower catalytic loading (2 mol%) had a positive effect on the reaction (Table 1, entries 7–10). Other solvents, including toluene and chlorobenzene, could better promote this transformation and then utilize chlorobenzene for further optimization (Table 1, entries 11–16). A further variation of reaction temperatures revealed that 80 °C was the optimal condition (Table 1, entries 17–19). The reaction time extension did not benefit the product yield (Table 1, entries 20–22). Thus, the optimal reaction conditions were Rh_2_(oct)_4_ in chlorobenzene at 90 °C for 12 h (Table 1, entry 13).

With the optimal conditions in hand, we explored the scope and generality of this [3 + 2] annulation with a combination of various substituted *N*-sulfonyl-1,2,3-triazoles **1** and *β*-enamino ketones **2** (Figure 2). We first evaluated the effect of substituents in the R^1^ group on the phenyl of *N*-sulfonyl-1,2,3-triazoles. The results indicated that the introduction of electron-neutral (-Me, -Et), electron-rich (-OMe), and electron-deficient (-F, -Cl, -Br) substituents at the *para*-positions was tolerated in this transformation. The desired imidazoles (products **3b**–**3g**) were obtained in yields ranging from 91% to 96%. Notably, the presence of bulky *tert*-butyl or strong electron-withdrawing trifluoromethyl groups at the para-position of benzene ring triazoles **1** led to a smooth reaction process. This resulted in the formation of the corresponding products **3h** and **3i**, with yields of 93% and 75%, respectively. Moreover, the extended π structure did not show an influence, and the desired product **3j** was successfully obtained with an 80% yield. Additionally, substituent variations on the *meta*- and *ortho*-positions could work well to produce the corresponding products **3k**–**3n** in 79–95% yields. Furthermore, the *N*-arylsulfonyl groups of the triazole substrates were also examined. The reactions of fluoro- and bromo-substituted phenylsulfonyl triazoles proceeded well, giving the desired products **3o** and **3p** in 91% and 76% yields, respectively. In addition, (*Z*)-3-amino-1-phenylpent-2-en-1-one was also a viable substrate for the transformation, generating the product **3q** in 56% yield.

Subsequently, an unexpected pyrrole product **5a** was obtained in 91% yield under standard conditions when the phenyl group (**4a**) replaced the methyl group of *β*-enaminones. We further evaluated the feasibility by using the 1,3-diaryl *β*-enaminones as starting materials (Figure 3). As expected, a wide range of electronically different substituents, including alkyl, methoxy, halogen, and bulky *tert*-butyl groups, were successfully installed into the products **5a**–**5h**. Moreover, an extended π-system was implemented on the pyrrole structure (product **5i**). Particularly noteworthy is that the halogen groups (e.g., -F, -Cl, and -Br) remained intact during the course of the reaction, which makes this transformation particularly attractive in terms of increasing the molecular complexity via transition metal-catalyzed coupling reactions (**5e**–**5g** and **5k**–**5m**). Additionally, we turned our attention to investigating the suitability of the substrate 1,3-diaryl *β*-enaminones **4**, and the desired products **5n**–**5q** were successfully obtained in 76–92% yields. It was gratifying that the introduction of a naphthyl and thienyl group also proceeded smoothly to produce the desired products **5o** and **5p** in yields of 92% and 90%, respectively. Likewise, changing the phenyl group to a bulky isopropyl was also tolerated in the reaction to give the desired product **5q** in an 86% yield.

Based on the above results, we hypothesize that the reaction of *N*-sulfonyl-1,2,3-triazoles and *β*-enaminones might be controlled by the steric hindrance of the substituent on the α-position of amino. Under standard conditions, when a moderate steric group such as *n*-propyl or *n*-butyl was present at the amino α-position of *β*-enaminones, the reaction resulted in the formation of the corresponding products, namely imidazoles (**3r** and **3s**) and pyrroles (**5s** and **5t**), as depicted in Figure 4a. In the synthesis of pyrroles, the presence of a methyl *p*-tolyl on the amino group resulted in a satisfactory yield of compounds **7a** and **7b** (72% and 85%, respectively; Figure 4b).

Notably, the reaction could be easily scaled up. As shown in Figure 5, imidazole **3a** could be obtained with a satisfactory yield of 77% (1.44 g) when the scale of the reaction was increased to 5 mmol. Additionally, the 3,5-disubstituted pyrrole **5a** was obtained in a 75% yield (1.21 g) on the same scale. Subsequently, several transformations were performed to demonstrate the utility of the target products. The desulfonation of imidazole **3a** afforded the unprotected imidazole **8** in 96% yield (Figure 5a). In addition, treating pyrrole **5a** with hydroxylamine hydrochloride and iodomethane successfully realized the formation of pyrrolyl oxime **9** in 94% yield (Figure 5b). In the presence of sodium hydride, *N*-methylation between compound **5a** and iodomethane easily generated *N*-methylpyrrole derivative **10** in 95% yield (Figure 5c). This highlights the synthetic utility of the current protocol.

The mechanism of this reaction was proposed as shown in Figure 6 based on the above experimental results and previous reports [40,41,42]. *α*-Diazo imino intermediate **A**, which was generated from the ring-chain tautomerization of triazole **1a**, could be efficiently decomposed by the rhodium(II) catalyst to form *α*-imino rhodium carbene intermediate **B** along with the release of nitrogen gas. *β*-Enaminones (**2a** or **4a**) attacked the electrophilic carbene center of intermediate **B**, and 1,1-insertion occurred to convert intermediate **D** with the rhodium(II) catalyst regeneration. In the case where R was a methyl group, an imino–enamine tautomerization could be triggered, leading to the formation of a more stable intermediate **E**. This intermediate **E** then underwent an intramolecular 1,4-conjugate addition, resulting in the formation of intermediate **F**. The elimination of the intermediate **F** results in the desired product **3a**. In the case of **4a**, the phenyl group was bulky enough to form the intermediate **D’**. Therefore, after the subsequent intramolecular nucleophilic addition and elimination processes, the corresponding product **5a** was obtained.

## 3. Materials and Methods

Unless otherwise specified, all reagents and starting materials were purchased from commercial sources and used as received. The solvents were purified and dried using standard procedures. The chromatography solvents were technical grade and distilled prior to use. The NMR spectra were recorded with a Bruker Avance 500 spectrometer (500 MHz for ^1^H and 125 MHz for ^13^C) with CDCl_3_ as a solvent and tetramethylsilane (TMS) as the internal standard at room temperature. Chemical shifts are given in *δ* relative to TMS, and the coupling constants *J* are given in Hz (Appendix A: ^1^H NMR and ^13^C NMR). HRMS spectra were obtained with an Agilent 6200 using a quadrupole time-of-flight mass spectrometer equipped with an ESI source. The melting points were measured using the SGWX-4 melting point apparatus and were not corrected. The X-ray source used for the single crystal X-ray diffraction analysis of compounds **3a** and **5a** was Mo Kα (λ = 0.71073 Å), and the thermal ellipsoid was drawn at the 30% probability level (Appendix A: X-ray crystal data).

### 3.1. General Procedure for the Synthesis of Trisubstituted Imidazoles ***3*** and Pyrroles ***5***

*N*-Sulfonyl-1H-1,2,3-triazoles **1** (0.2 mmol), *β*-enaminones **2** (0.2 mmol), and Rh_2_(oct)_4_ (2 mol%) were successively added to a Schlenk reaction tube. The reaction set was evacuated and backfilled with argon three times. Then, chlorobenzene (2.0 mL) was added to the reaction tube through a syringe. The reaction mixture was stirred vigorously in an oil bath preheated to 90 °C for 12 h. After the reaction was complete, the reaction mixture was cooled to room temperature, extracted with CH_2_Cl_2_ (3 × 10 mL), and washed with brine. The organic layers were combined, dried over Na_2_SO_4_, and then evaporated under a vacuum. The residue was purified by flash column chromatography on silica gel (200–300 mesh) using ethyl acetate and petroleum ether (1:8, *v*/*v*) as the elution solvents to give desired products **3** or **5**.

### 3.2. General Procedure for the Synthesis of Compound ***8***

2-Methyl-4-phenyl-1-tosyl-1*H*-imidazole **3a** (0.15 mmol) and NaOH (2.25 mmol) were successively added to a Schlenk reaction tube. The reaction set was evacuated and backfilled with argon three times. Then, methanol (2.0 mL) was added into the reaction tube through a syringe. The reaction mixture was stirred vigorously in an oil bath preheated to 70 °C for 30 min. After the reaction was complete, the reaction mixture was cooled to room temperature, extracted with CH_2_Cl_2_ (3 × 10 mL), and washed with brine. The organic layers were combined, dried over Na_2_SO_4_, and then evaporated under a vacuum. The residue was purified by flash column chromatography on silica gel (200–300 mesh) using ethyl acetate and petroleum ether (1:3, *v/v*) as the elution solvents to give the desired product **8** in a 96% yield.

### 3.3. General Procedure for the Synthesis of Compound ***9***

A mixture of (2,5-diphenyl-1*H*-pyrrol-3-yl)(phenyl)methanone **5a** (0.2 mmol), hydroxylamine hydrochloride (0.4 mmol), and sodium acetate (0.5 mmol) was added to a round-bottomed flask with a reflux condenser. Ethanol (4 mL) was then added, and the reaction mixture was stirred vigorously at reflux in an oil bath for 12 h. After quenching with water, the residue was extracted twice with ethyl acetate. The combined layer was washed with brine, dried over Na_2_SO_4_, and then evaporated under a vacuum. The residue was purified by flash column chromatography on silica gel (200–300 mesh) using ethyl acetate and petroleum ether (1:8, *v/v*) as the elution solvents to give the desired product **9** in a 94% yield.

### 3.4. General Procedure for the Synthesis of Compound ***10***

NaH (60% in mineral oil, 0.5 mmol, 1.7 equiv.) was added to a solution of **5a** (0.25 mmol) in DCM (4 mL) at 0 °C in portions. After stirring for 5 min at 0 °C, MeI (0.22 mmol, 1.1 equiv.) was added dropwise, and the reaction mixture was allowed to warm to room temperature and stirred for another 19 h. After quenching with water, the residue was extracted twice with ethyl acetate. The combined organic layer was washed with brine, dried over Na_2_SO_4_, filtrated and concentrated, and purified by column chromatography to afford **10** in 95% yield.

*2-Methyl-4-phenyl-1-tosyl-1H-imidazole* (**3a**). This compound was purified by column chromatography (ethyl acetate/petroleum ether = 1:8) to afford a white solid in a 96% yield (60 mg); mp 122–124 °C; ^1^H NMR (500 MHz, CDCl_3_) *δ* 7.81 (d, *J* = 8.5 Hz, 2H), 7.73 (d, *J* = 7.0 Hz, 2H), 7.67 (s, 1H), 7.39–7.35 (m, 4H), 7.25 (d, *J* = 7.5 Hz, 1H), 2.57 (s, 3H), 2.44 (s, 3H); ^13^C NMR (125 MHz, CDCl_3_) *δ* 146.5, 146.4, 141.0, 135.5, 132.7, 130.8, 129.1, 128.2, 127.8, 125.6, 114.4, 22.1, 15.6; HRMS (ESI-TOF) *m/z*: [M + H]^+^ Calcd for C_17_H_17_N_2_O_2_S 313.1005; found 313.1006.*2-Methyl-4-(p-tolyl)-1-tosyl-1H-imidazole* (**3b**). This compound was purified by column chromatography (ethyl acetate/petroleum ether = 1:8) to afford a white solid in a 96% yield (62 mg); mp 60–62 °C; ^1^H NMR (500 MHz, CDCl_3_) *δ* 7.80 (d, *J* = 8.5 Hz, 2H), 7.62 (d, *J* = 8.0 Hz, 3H), 7.35 (d, *J* = 8.0 Hz, 2H), 7.18 (d, *J* = 8.0 Hz, 2H), 2.57 (s, 3H), 2.44 (s, 3H), 2.35 (s, 3H); ^13^C NMR (125 MHz, CDCl_3_) *δ* 146.4, 146.3, 141.0, 138.0, 135.4, 130.7, 129.9, 129.8, 127.7, 125.5, 113.9, 22.1, 21.7, 15.6; HRMS (ESI-TOF) *m/z*: [M + H]^+^ Calcd for C_18_H_19_N_2_O_2_S 327.1162; found 327.1170.*4-(4-Ethylphenyl)-2-methyl-1-tosyl-1H-imidazole* (**3c**). This compound was purified by column chromatography (ethyl acetate/petroleum ether = 1:8) to afford a white solid in a 95% yield (64 mg); mp 77–79 °C; ^1^H NMR (500 MHz, CDCl_3_) *δ* 7.80 (d, *J* = 8.5 Hz, 2H), 7.65 (d, *J* = 8.0 Hz, 3H), 7.34 (d, *J* = 8.0 Hz, 2H), 7.21 (d, *J* = 8.0 Hz, 2H), 2.64 (q, *J* = 7.5 Hz, 2H), 2.57 (s, 3H), 2.42 (s, 3H), 1.23 (t, *J* = 7.5 Hz, 3H); ^13^C NMR (125 MHz, CDCl_3_) *δ* 146.4, 146.3, 144.4, 141.0, 135.4, 130.8, 130.1, 128.6, 127.7, 125.5, 113.9, 29.1, 22.1, 15.9, 15.6; HRMS (ESI-TOF) *m/z*: [M + H]^+^ Calcd for C_19_H_21_N_2_O_2_S 341.1318; found 341.1319.*4-(4-Methoxyphenyl)-2-methyl-1-tosyl-1H-imidazole* (**3d**). This compound was purified by column chromatography (ethyl acetate/petroleum ether = 1:8) to afford a white solid in a 91% yield (62 mg); mp 66–68 °C; ^1^H NMR (500 MHz, CDCl_3_) *δ* 7.80 (d, *J* = 8.5 Hz, 2H), 7.65 (d, *J* = 9.0 Hz, 2H), 7.57 (s, 1H), 7.34 (d, *J* = 8.0 Hz, 2H), 6.90 (d, *J* = 9.0 Hz, 2H), 3.81 (s, 3H), 2.56 (s, 3H), 2.42 (s, 3H); ^13^C NMR (125 MHz, CDCl_3_) *δ* 159.8, 146.4, 146.3, 140.8, 135.4, 130.7, 127.7, 126.9, 125.5, 114.5, 113.2, 55.7, 22.1, 15.6; HRMS (ESI-TOF) *m/z*: [M + H]^+^ Calcd for C_18_H_19_N_2_O_3_S 343.1111; found 343.1112.*4-(4-Fluorophenyl)-2-methyl-1-tosyl-1H-imidazole* (**3e**). This compound was purified by column chromatography (ethyl acetate/petroleum ether = 1:8) to afford a white solid in a 94% yield (62 mg); mp 107–109 °C; ^1^H NMR (500 MHz, CDCl_3_) *δ* 7.81 (d, *J* = 8.5 Hz, 2H), 7.69 (dd, *J* = 8.5, 5.0 Hz, 2H), 7.61 (s, 1H), 7.36 (d, *J* = 8.0 Hz, 2H), 7.06 (t, *J* = 8.5 Hz, 2H), 2.56 (s, 3H), 2.44 (s, 3H); ^13^C NMR (125 MHz, CDCl_3_) *δ* 162.9 (d, *J_C-F_* = 246.3 Hz), 146.6, 146.5, 140.1, 135.3, 130.8, 129.0, 127.8, 127.3 (d, *J_C-F_* = 8.0 Hz), 116.0 (d, *J_C-F_* = 21.6 Hz), 114.0, 22.1, 15.6; HRMS (ESI-TOF) *m/z*: [M + H]^+^ Calcd for C_17_H_16_FN_2_O_2_S 331.0911; found 331.0909.*4-(4-Chlorophenyl)-2-methyl-1-tosyl-1H-imidazole* (**3f**). This compound was purified by column chromatography (ethyl acetate/petroleum ether = 1:8) to afford a white solid in a 94% yield (65 mg); mp 107–109 °C; ^1^H NMR (500 MHz, CDCl_3_) *δ* 7.80 (d, *J* = 7.5 Hz, 2H), 7.66–7.65 (m, 3H), 7.36–7.32 (m, 4H), 2.56 (s, 3H), 2.43 (s, 3H); ^13^C NMR (125 MHz, CDCl_3_) *δ* 146.6, 146.6, 139.9, 135.2, 133.8, 131.3, 130.8, 129.2, 127.8, 126.8, 114.5, 22.1, 15.5; HRMS (ESI-TOF) *m/z*: [M + H]^+^ Calcd for C_17_H_16_ClN_2_O_2_S 347.0616; found 347.0612.*4-(4-Bromophenyl)-2-methyl-1-tosyl-1H-imidazole* (**3g**). This compound was purified by column chromatography (ethyl acetate/petroleum ether = 1:8) to afford a white solid in a 93% yield (72 mg); mp 104–106 °C; ^1^H NMR (500 MHz, CDCl_3_) *δ* 7.80 (d, *J* = 8.5 Hz, 2H), 7.67 (s, 1H), 7.60 (d, *J* = 8.5 Hz, 2H), 7.48 (d, *J* = 8.5 Hz, 2H), 7.35 (d, *J* = 8.5 Hz, 2H), 2.56 (s, 3H), 2.42 (s, 3H); ^13^C NMR (125 MHz, CDCl_3_) *δ* 146.7, 146.6, 139.9, 135.2, 132.2, 131.7, 130.8, 127.8, 127.1, 122.0, 114.6, 22.1, 15.6; HRMS (ESI-TOF) *m/z*: [M + H]^+^ Calcd for C_17_H_16_BrN_2_O_2_S 391.0110; found 391.0109.*4-(4-(tert-Butyl)phenyl)-2-methyl-1-tosyl-1H-imidazole* (**3h**). This compound was purified by column chromatography (ethyl acetate/petroleum ether = 1:8) to afford a white solid in a 93% yield (68 mg); mp 69–71 °C; ^1^H NMR (500 MHz, CDCl_3_) *δ* 7.79 (d, *J* = 7.0 Hz, 2H), 7.69–7.64 (m, 3H), 7.39 (d, *J* = 7.0 Hz, 2H), 7.34 (d, *J* = 8.0 Hz, 2H), 2.57 (s, 3H), 2.42 (s, 3H), 1.32 (s, 9H); ^13^C NMR (125 MHz, CDCl_3_) *δ* 151.3, 146.4, 146.4, 141.0, 135.5, 130.7, 129.9, 127.7, 126.0, 125.3, 114.0, 35.0, 31.7, 22.1, 15.6; HRMS (ESI-TOF) *m/z*: [M + H]^+^ Calcd for C_21_H_25_N_2_O_2_S 369.1631; found 369.1634.*2-Methyl-1-tosyl-4-(4-(trifluoromethyl)phenyl)-1H-imidazole* (**3i**). This compound was purified by column chromatography (ethyl acetate/petroleum ether = 1:8) to afford a white solid in a 75% yield (57 mg); mp 76–78 °C; ^1^H NMR (500 MHz, CDCl_3_) *δ* 7.84–7.82 (m, 4H), 7.76 (s, 1H), 7.62 (d, *J* = 8.0 Hz, 2H), 7.38 (d, *J* = 8.0 Hz, 2H), 2.58 (s, 3H), 2.45 (s, 3H); ^13^C NMR (125 MHz, CDCl_3_) *δ* 146.8, 139.5, 136.2, 135.1, 130.9, 129.9 (q, *J_C-F_* = 32.5 Hz), 128.5, 127.9, 126.6, 126.1(q, *J_C-F_* = 3.8 Hz), 124.6(q, *J_C-F_* = 270.0 Hz), 115.6, 22.1, 15.5; HRMS (ESI-TOF) *m/z*: [M + H] ^+^ Calcd for C_18_H_16_F_3_N_2_O_2_S 381.0879; found 381.0878.*2-Methyl-4-(4’-propyl-[1,1’-biphenyl]-4-yl)-1-tosyl-1H-imidazole* (**3j**). This compound was purified by column chromatography (ethyl acetate/petroleum ether = 1:8) to afford a white solid in an 80% yield (69 mg); mp 94–96 °C; ^1^H NMR (500 MHz, CDCl_3_) *δ* 7.83–7.79 (m, 4H), 7.71(s, 1H), 7.61 (d, *J* = 8.5 Hz, 2H), 7.54 (d, *J* = 8.0 Hz, 2H), 7.36 (d, *J* = 8.0 Hz, 2H), 7.26 (d, *J* = 8.5 Hz, 2H), 2.63 (t, *J* = 7.5 Hz, 2H), 2.59 (s, 3H), 1.72–1.64 (m, 2H), 0.98 (t, *J* = 7.5 Hz, 3H); ^13^C NMR (125 MHz, CDCl_3_) *δ* 146.5, 146.5, 142.4, 140.9, 140.7, 138.4, 135.4, 131.4, 130.8, 129.3, 127.8, 127.6, 127.1, 125.9, 114.3, 38.1, 25.0, 22.1, 15.6, 14.3; HRMS (ESI-TOF) *m/z*: [M + H]^+^ Calcd for C_26_H_27_N_2_O_2_S 431.1788; found 431.1781.*2-Methyl-4-(m-tolyl)-1-tosyl-1H-imidazole* (**3k**). This compound was purified by column chromatography (ethyl acetate/petroleum ether = 1:8) to afford a white solid in a 95% yield (62 mg); mp 107–109 °C; ^1^H NMR (500 MHz, CDCl_3_) *δ* 7.80 (d, *J* = 8.5 Hz, 2H), 7.67 (s, 1H), 7.58 (s, 1H), 7.51 (d, *J* = 7.5 Hz, 1H), 7.35 (d, *J* = 8.0 Hz, 2H), 7.26 (t, *J* = 7.5 Hz, 1H), 7.09 (d, *J* = 7.5 Hz, 1H), 2.57 (s, 3H), 2.43 (s, 3H), 2.37 (s, 3H); ^13^C NMR (125 MHz, CDCl_3_) *δ* 146.5, 146.4, 141.0, 138.8, 135.4, 132.5, 130.8, 129.0, 127.8, 126.2, 122.6, 114.3, 22.1, 21.8, 15.6; HRMS (ESI-TOF) *m/z*: [M + H]^+^ Calcd for C_18_H_19_N_2_O_2_S 327.1162; found 327.1163.*4-(3-Chlorophenyl)-2-methyl-1-tosyl-1H-imidazole* (**3l**). This compound was purified by column chromatography (ethyl acetate/petroleum ether = 1:8) to afford a white solid in an 85% yield (59 mg); mp 83–85 °C; ^1^H NMR (500 MHz, CDCl_3_) *δ* 7.81 (d, *J* = 8.0 Hz, 2H), 7.73 (s, 1H), 7.68 (s, 1H), 7.59 (d, *J* = 7.5 Hz, 1H), 7.37 (d, *J* = 8.5 Hz, 2H), 7.29 (t, *J* = 8.0 Hz, 1H), 7.24 (d, *J* = 8.0 Hz, 1H), 2.56 (s, 3H), 2.44 (s, 3H); ^13^C NMR (125 MHz, CDCl_3_) *δ* 146.7, 139.7, 135.1, 134.6, 130.8, 130.3, 128.1, 127.8, 125.7, 123.6, 115.0, 22.1, 15.5; HRMS (ESI-TOF) *m/z*: [M + H]^+^ Calcd for C_17_H_16_ClN_2_O_2_S 347.0616; found 347.0625.*4-(3-Bromophenyl)-2-methyl-1-tosyl-1H-imidazole* (**3m**). This compound was purified by column chromatography (ethyl acetate/petroleum ether = 1:8) to afford a white solid in an 82% yield (64 mg); mp 79–81 °C; ^1^H NMR (500 MHz, CDCl_3_) *δ* 7.89 (s, 1H), 7.80 (d, *J* = 8.5 Hz, 2H), 7.67 (s, 1H), 7.64 (d, *J* = 7.5 Hz, 1H), 7.39–7.35 (m, 3H), 7.22 (t, *J* = 8.0 Hz, 1H), 2.56 (s, 3H), 2.43 (s, 3H); ^13^C NMR (125 MHz, CDCl_3_) *δ* 146.7, 146.6, 139.5, 135.2, 134.8, 131.0, 130.8, 130.6, 128.6, 127.8, 124.1, 123.3, 115.0, 22.1, 15.5; HRMS (ESI-TOF) *m/z*: [M + H]^+^ Calcd for C_17_H_16_BrN_2_O_2_S 391.0110; found 391.0119.*4-(2-Fluorophenyl)-2-methyl-1-tosyl-1H-imidazole* (**3n**). This compound was purified by column chromatography (ethyl acetate/petroleum ether = 1:8) to afford a white solid in a 79% yield (52 mg); mp 57–59 °C; ^1^H NMR (500 MHz, CDCl_3_) *δ* 8.03 (t, *J* = 7.5 Hz, 1H), 7.85 (d, *J* = 4.0 Hz, 1H), 7.82 (d, *J* = 8.0 Hz, 2H), 7.35 (d, *J* = 8.0 Hz, 2H), 7.23 (t, *J* = 7.0 Hz, 1H), 7.17 (t, *J* = 7.5 Hz, 1H), 7.10 (t, *J* = 10 Hz, 1H), 2.58 (s, 3H), 2.43 (s, 3H); ^13^C NMR (125 MHz, CDCl_3_) *δ* 160.2 (d, *J_C-F_* = 247.9 Hz), 146.5, 145.9, 135.4, 134.5, 130.8, 129.1 (d, *J_C-F_* = 8.5 Hz), 128.2 (d, *J_C-F_* = 3.6 Hz), 127.8, 124.7 (d, *J_C-F_* = 3.6 Hz), 120.6 (d, *J_C-F_* = 12.5 Hz), 118.4 (d, *J_C-F_* = 15.4 Hz), 116.0 (d, *J_C-F_* = 21.5 Hz), 22.1, 15.5; HRMS (ESI-TOF) *m/z*: [M + H]^+^ Calcd for C_17_H_16_FN_2_O_2_S 331.0911; found 331.0914.*1-((4-Fluorophenyl)sulfonyl)-2-methyl-4-phenyl-1H-imidazole* (**3o**). This compound was purified by column chromatography (ethyl acetate/petroleum ether = 1:8) to afford a white solid in a 91% yield (57 mg); mp 107–109 °C; ^1^H NMR (500 MHz, CDCl_3_) *δ* 7.97–7.94 (m, 2H), 7.73 (d, *J* = 7.5 Hz, 2H), 7.67 (s, 1H), 7.38 (t, *J* = 8.0 Hz, 2H), 7.29 (t, *J* = 7.5 Hz, 1H), 7.24 (t, *J* = 8.0 Hz, 2H), 2.58 (s, 3H); ^13^C NMR (125 MHz, CDCl_3_) *δ* 166.6 (d, *J_C-F_* = 257.5 Hz), 146.4, 141.3, 134.4 (d, *J_C-F_* = 2.8 Hz), 132.5, 130.7 (d, *J_C-F_* = 9.8 Hz), 129.1, 128.4, 125.6, 117.7 (d, *J_C-F_* = 22.9 Hz), 114.2, 15.7; HRMS (ESI-TOF) *m/z*: [M + H]^+^ Calcd for C_16_H_14_FN_2_O_2_S 317.0755; found 317.0760.*1-((4-Bromophenyl)sulfonyl)-2-methyl-4-phenyl-1H-imidazole* (**3p**). This compound was purified by column chromatography (ethyl acetate/petroleum ether = 1:8) to afford a white solid in a 76% yield (57 mg); mp 97–99 ℃; ^1^H NMR (500 MHz, CDCl_3_) *δ* 7.78 (d, *J* = 9.0 Hz, 2H), 7.72 (t, *J* = 9.0 Hz, 4H), 7.65 (s, 1H), 7.38 (d, *J* = 7.5 Hz, 2H), 7.29 (d, *J* = 7.5 Hz, 1H), 2.58 (s, 3H); ^13^C NMR (125 MHz, CDCl_3_) *δ* 146.5, 141.4, 137.3, 133.6, 132.4, 130.6, 129.1, 129.1, 128.4, 125.6, 114.2, 15.7; HRMS (ESI-TOF) *m/z*: [M + H]^+^ Calcd for C_16_H_14_BrN_2_O_2_S 376.9954; found 376.9952.*2-Ethyl-4-phenyl-1-tosyl-1H-imidazole* (**3q**). This compound was purified by column chromatography (ethyl acetate/petroleum ether = 1:8) to afford a white solid in a 56% yield (37 mg); mp 49–51 °C; ^1^H NMR (500 MHz, CDCl_3_) *δ* 7.79 (d, *J* = 8.5 Hz, 2H), 7.75 (d, *J* = 7.0 Hz, 2H), 7.67 (s, 1H), 7.39-7.34 (m, 4H), 7.28 (d, *J* = 7.5 Hz, 1H), 2.90 (q, *J* = 7.5 Hz, 2H), 2.43 (s, 3H), 1.32 (t, *J* = 7.5 Hz, 3H); ^13^C NMR (125 MHz, CDCl_3_) *δ* 151.4, 146.4, 140.9, 135.7, 132.9, 130.7, 129.0, 128.1, 127.7, 125.6, 114.3, 22.4, 22.1, 12.5; HRMS (ESI-TOF) *m/z*: [M + H]^+^ Calcd for C_18_H_19_N_2_O_2_S 327.1162; found 327.1163.*4-Phenyl-2-propyl-1-tosyl-1H-imidazole* (**3r**). This compound was purified by column chromatography (ethyl acetate/petroleum ether = 1:8) to afford a colorless oil in a 45% yield (31 mg); ^1^H NMR (500 MHz, CDCl_3_) *δ* 7.79 (d, *J* = 8.5 Hz, 2H), 7.75 (d, *J* = 7.0 Hz, 2H), 7.66 (s, 1H), 7.38–7.33 (m, 4H), 7.27 (t, *J* = 7.5 Hz, 1H), 2.85 (t, *J* = 7.5 Hz, 2H), 2.43 (s, 3H), 1.81–1.74 (m, 2H), 0.98 (t, *J* = 7.5 Hz, 3H); ^13^C NMR (125 MHz, CDCl_3_) *δ* 150.4, 146.4, 141.0, 135.8, 132.9, 130.7, 129.0, 128.1, 127.6, 125.6, 114.3, 30.8, 22.1, 21.9, 14.3; HRMS (ESI-TOF) *m/z*: [M + H]^+^ Calcd for C_19_H_21_N_2_O_2_S 341.1318; found 341.1312.*2-Butyl-4-phenyl-1-tosyl-1H-imidazole* (**3s**). This compound was purified by column chromatography (ethyl acetate/petroleum ether = 1:8) to afford a colorless oil in a 44% yield (31 mg); ^1^H NMR (500 MHz, CDCl_3_) *δ* 7.79 (d, *J* = 8.5 Hz, 2H), 7.75 (d, *J* = 7.5 Hz, 2H), 7.67 (s, 1H), 7.38–7.33 (m, 4H), 7.28 (d, *J* = 7.5 Hz, 1H), 2.87 (t, *J* = 8 Hz, 2H), 2.43 (s, 3H), 1.72–1.69 (m, 2H), 1.42–1.37 (m, 2H), 0.91 (t, *J* = 7.5 Hz, 3H); ^13^C NMR (125 MHz, CDCl_3_) *δ* 150.5, 146.4, 140.9, 135.7, 132.9, 130.7, 129.1, 128.2, 127.7, 125.6, 114.3, 30.5, 28.6, 22.9, 22.1, 14.2; HRMS (ESI-TOF) *m/z*: [M + H]^+^ Calcd for C_20_H_23_N_2_O_2_S 355.1475; found 355.1482.*(2,5-Diphenyl-1H-pyrrol-3-yl)(phenyl)methanone* (**5a**). This compound was purified by column chromatography (ethyl acetate/petroleum ether = 1:8) to afford a yellow solid in a 91% yield (59 mg); mp 81–83 °C; ^1^H NMR (500 MHz, CDCl_3_) *δ* 8.98 (s, 1H), 7.80 (d, *J* = 7.0 Hz, 2H), 7.54 (d, *J* = 7.5 Hz, 2H), 7.45-7.42 (m, 3H), 7.39 (t, *J* = 7.5 Hz, 2H), 7.32 (t, *J* = 7.5 Hz, 2H), 7.29–7.24 (m, 4H), 6.84 (d, *J* = 3.0 Hz, 1H); ^13^C NMR (125 MHz, CDCl_3_) *δ* 192.9, 139.8, 138.3, 132.3, 132.1, 131.8, 130.1, 129.5, 128.9, 128.8, 128.5, 128.3, 127.5, 124.5, 122.3, 110.9; HRMS (ESI-TOF) *m/z*: [M + H]^+^ Calcd for C_23_H_18_NO 324.1383; found 324.1382.*Phenyl(2-phenyl-5-(p-tolyl)-1H-pyrrol-3-yl)methanone* (**5b**). This compound was purified by column chromatography (ethyl acetate/petroleum ether = 1:8) to afford a yellow solid in a 91% yield (61 mg); mp 81–83 °C; ^1^H NMR (500 MHz, CDCl_3_) *δ* 9.12 (s, 1H), 7.79 (d, *J* = 8.0 Hz, 2H), 7.47–7.38 (m, 5H), 7.31 (t, *J* = 7.7 Hz, 2H), 7.21–7.17 (m, 5H), 6.78 (s, 1H), 2.36 (s, 3H); ^13^C NMR (125 MHz, CDCl_3_) *δ* 193.1, 139.8, 138.1, 137.3, 132.5, 132.2, 132.1, 130.1, 130.1, 129.1, 128.9, 128.7, 128.4, 128.3, 124.6, 122.2, 110.4, 21.6; HRMS (ESI-TOF) *m/z*: [M + H]^+^ Calcd for C_24_H_20_NO 338.1539; found 338.1545.*(5-(4-Ethylphenyl)-2-phenyl-1H-pyrrol-3-yl)(phenyl)methanone* (**5c**). This compound was purified by column chromatography (ethyl acetate/petroleum ether = 1:8) to afford a yellow solid in a 92% yield (64 mg); mp 71–73 °C; ^1^H NMR (500 MHz, CDCl_3_) *δ* 8.92 (s, 1H), 7.80 (d, *J* = 7.0 Hz, 2H), 7.47-7.42 (m, 5H), 7.32 (t, *J* = 7.5 Hz, 2H), 7.26–7.22 (m, 5H), 6.81 (d, *J* = 3.0 Hz, 1H), 2.67 (q, *J* = 7.5 Hz, 2H), 1.26 (t, *J* = 7.5 Hz, 3H); ^13^C NMR (125 MHz, CDCl_3_) *δ* 193.0, 143.8, 139.8, 138.0, 132.5, 132.2, 132.1, 130.1, 129.3, 128.9, 128.9, 128.7, 128.4, 128.3, 124.6, 122.2, 110.4, 29.0, 15.9; HRMS (ESI-TOF) *m/z*: [M + H]^+^ Calcd for C_25_H_22_NO 352.1693; found 352.1689.*(5-(4-Methoxyphenyl)-2-phenyl-1H-pyrrol-3-yl)(phenyl)methanone* (**5d**). This compound was purified by column chromatography (ethyl acetate/petroleum ether = 1:8) to afford a yellow solid in a 92% yield (65 mg); mp 111–113 °C; ^1^H NMR (500 MHz, CDCl_3_) *δ* 8.96 (s, 1H), 7.79 (d, *J* = 7.5 Hz, 2H), 7.46 (d, *J* = 8.5 Hz, 2H), 7.41 (d, *J* = 7.0 Hz, 3H), 7.31 (t, *J* = 7.5 Hz, 2H), 7.22 (d, *J* = 7.5 Hz, 3H), 6.92 (d, *J* = 8.5 Hz, 2H), 6.72 (d, *J* = 3.0 Hz, 1H), 3.82 (s, 3H); ^13^C NMR (125 MHz, CDCl_3_) *δ* 193.1, 159.3, 139.9, 137.9, 132.4, 132.3, 132.1, 130.1, 128.9, 128.7, 128.3, 128.3, 126.0, 124.8, 122.2, 114.9, 109.8, 55.8; HRMS (ESI-TOF) *m/z*: [M + H]^+^ Calcd for C_24_H_20_NO 354.1489; found 354.1489.*(5-(4-Fluorophenyl)-2-phenyl-1H-pyrrol-3-yl)(phenyl)methanone* (**5e**). This compound was purified by column chromatography (ethyl acetate/petroleum ether = 1:8) to afford a yellow solid in an 85% yield (58 mg); mp 104–106 °C; ^1^H NMR (500 MHz, CDCl_3_) *δ* 8.93 (s, 1H), 7.78 (d, *J* = 7.0 Hz, 2H), 7.50 (dd, *J* = 9.0, 5.0 Hz, 2H), 7.45–7.41 (m, 3H), 7.31 (t, *J* = 7.5 Hz, 2H), 7.24–7.22 (m, 3H), 7.08 (t, *J* = 8.5 Hz, 2H), 6.76 (d, *J* = 7.5 Hz, 1H); ^13^C NMR (125 MHz, CDCl_3_) *δ* 192.9, 162.4 (d, *J_C-F_* = 245.0 Hz), 139.7, 138.3, 132.2, 132.1, 131.5, 130.1, 128.8 (d, *J_C-F_* = 6.3 Hz), 128.6, 128.3, 128.2, 126.4, 126.3, 122.4, 116.5 (d, *J_C-F_* = 22.5 Hz), 110.7; HRMS (ESI-TOF) *m/z*: [M + H]^+^ Calcd for C_23_H_17_FNO 342.1289; found 342.1287.*(5-(4-Chlorophenyl)-2-phenyl-1H-pyrrol-3-yl)(phenyl)methanone* (**5f**). This compound was purified by column chromatography (ethyl acetate/petroleum ether = 1:8) to afford a yellow solid in an 84% yield (60 mg); mp 112–114 ℃; ^1^H NMR (500 MHz, CDCl_3_) *δ* 9.04 (s, 1H), 7.77 (d, *J* = 7.0Hz, 2H), 7.46–7.42 (m, 3H), 7.39 (dd, *J* = 6.5, 3.0 Hz, 2H), 7.35–7.30 (m, 4H), 7.23–7.19 (m, 3H), 6.79 (d, *J* = 3.0 Hz, 1H); ^13^C NMR (125 MHz, CDCl_3_) *δ* 192.9, 139.6, 138.6, 133.1, 132.2, 131.9, 131.3, 130.4, 130.1, 129.6, 128.9, 128.8, 128.6, 128.3, 125.8, 122.4, 111.2; HRMS (ESI-TOF) *m/z*: [M + H]^+^ Calcd for C_23_H_17_ClNO 358.0993; found 358.1002.*(5-(4-Bromophenyl)-2-phenyl-1H-pyrrol-3-yl)(phenyl)methanone* (**5g**). This compound was purified by column chromatography (ethyl acetate/petroleum ether = 1:8) to afford a yellow solid in an 84% yield (67 mg); mp 127–129 °C; ^1^H NMR (500 MHz, CDCl_3_) *δ* 8.77 (s, 1H), 7.79 (d, *J* = 8.5 Hz, 2H), 7.52 (d, *J* = 8.5 Hz, 2H), 7.46–7.43 (m, 3H), 7.40 (d, *J* = 8.5 Hz, 2H), 7.33 (t, *J* = 7.5 Hz, 2H), 7.29-7.27 (m, 3H), 6.85 (d, *J* = 7.5 Hz, 1H); ^13^C NMR (125 MHz, CDCl_3_) *δ* 193.0, 139.6, 138.7, 132.5, 132.3, 131.9, 131.3, 130.8, 130.1, 128.9, 128.7, 128.6, 128.3, 126.1, 122.4, 121.1, 111.3; HRMS (ESI-TOF) *m/z*: [M + H]^+^ Calcd for C_23_H_17_BrNO 402.0488; found 402.0489.*(5-(4-(tert-Butyl)phenyl)-2-phenyl-1H-pyrrol-3-yl)(phenyl)methanone* (**5h**). This compound was purified by column chromatography (ethyl acetate/petroleum ether = 1:8) to afford a yellow solid in a 93% yield (70 mg); mp 98–100 °C; ^1^H NMR (500 MHz, CDCl_3_) *δ* 8.89 (s, 1H), 7.80 (d, *J* = 8.0 Hz, 2H), 7.49–7.40 (m, 7H), 7.32 (t, *J* = 8.0 Hz, 2H), 7.28–7.23 (m, 3H), 6.82 (s, 1H), 1.34 (s, 9H); ^13^C NMR (125 MHz, CDCl_3_) *δ* 192.9, 150.7, 139.8, 138.0, 132.4, 132.3, 132.1, 130.1, 129.1, 128.9, 128.8, 128.4, 128.3, 126.4, 124.3, 122.3, 110.5, 35.0, 31.7; HRMS (ESI-TOF) *m/z*: [M + H]^+^ Calcd for C_27_H_26_NO 380.2009; found 380.2008.*Phenyl(2-phenyl-5-(4’-propyl-[1,1’-biphenyl]-4-yl)-1H-pyrrol-3-yl)methanone* (**5i**). This compound was purified by column chromatography (ethyl acetate/petroleum ether = 1:8) to afford a yellow solid in an 85% yield (72 mg); mp 149–151 °C; ^1^H NMR (500 MHz, CDCl_3_) *δ* 9.43 (s, 1H), 7.80 (d, *J* = 7.5 Hz, 2H), 7.59 (s, 4H), 7.52 (d, *J* = 8.0 Hz, 2H), 7.44 (t, *J* = 7.5 Hz, 1H), 7.41–7.36 (m, 2H), 7.32 (t, *J* = 7.5 Hz, 2H), 7.25 (d, *J* = 8.0 Hz, 2H), 7.20–7.15 (m, 3H), 6.85 (d, *J* = 3.0 Hz, 1H), 2.64 (d, *J* = 8.0 Hz, 2H), 1.69 (m, 2H), 0.99 (t, *J* = 7.5 Hz, 3H); ^13^C NMR (125 MHz, CDCl_3_) *δ* 193.3, 142.5, 140.0, 139.8, 138.7, 138.2, 132.3, 132.2, 132.1, 130.5, 130.2, 129.4, 129.0, 128.7, 128.4, 128.3, 127.8, 127.1, 125.0, 122.3, 110.9, 38.1, 25.0, 14.3; HRMS (ESI-TOF) *m/z*: [M + H]^+^ Calcd for C_32_H_28_NO 442.2165; found 442.2170.*Phenyl(2-phenyl-5-(m-tolyl)-1H-pyrrol-3-yl)methanone* (**5j**). This compound was purified by column chromatography (ethyl acetate/petroleum ether = 1:8) to afford a yellow solid in an 86% yield (58 mg); mp 118–120 °C; ^1^H NMR (500 MHz, CDCl_3_) *δ* 9.08 (s, 1H), 7.80 (d, *J* = 7.5 Hz, 2H), 7.43 (d, *J* = 6.0 Hz, 3H), 7.37–7.31 (m, 4H), 7.27 (d, *J* = 7.5 Hz, 1H), 7.22 (t, *J* = 6.0 Hz, 3H), 7.08 (d, *J* = 7.5 Hz, 1H), 6.82 (d, *J* = 2.5 Hz, 1H), 2.38 (s, 3H); ^13^C NMR (125 MHz, CDCl_3_) *δ* 193.1, 139.8, 139.0, 138.4, 132.5, 132.2, 132.1, 131.8, 130.1, 129.3, 128.9, 128.7, 128.4, 128.3, 125.4, 122.2, 121.7, 110.8, 21.9; HRMS (ESI-TOF) *m/z*: [M + H]^+^ Calcd for C_24_H_20_NO 338.1539; found 338.1531.*(5-(3-Chlorophenyl)-2-phenyl-1H-pyrrol-3-yl)(phenyl)methanone* (**5k**). This compound was purified by column chromatography (ethyl acetate/petroleum ether = 1:8) to afford a yellow solid in an 85% yield (60 mg); mp 83–85 °C; ^1^H NMR (500 MHz, CDCl_3_) *δ* 9.19 (s, 1H), 7.78 (d, *J* = 7.5 Hz, 2H), 7.51 (s, 1H), 7.44 (t, *J* = 7.5 Hz, 1H), 7.41–7.38 (m, 3H), 7.34–7.27 (m, 3H), 7.23–7.18 (m, 4H), 6.80 (d, *J* = 2.5 Hz, 1H); ^13^C NMR (125 MHz, CDCl_3_) *δ* 193.0, 139.6, 138.9, 135.4, 133.6, 132.3, 131.8, 130.9, 130.6, 130.1, 128.9, 128.7, 128.6, 128.4, 127.3, 124.6, 122.6, 122.3, 111.7; HRMS (ESI-TOF) *m/z*: [M + H]^+^ Calcd for C_23_H_17_ClNO 358.0993; found 358.0992.*(5-(3-Bromophenyl)-2-phenyl-1H-pyrrol-3-yl)(phenyl)methanone* (**5l**). This compound was purified by column chromatography (ethyl acetate/petroleum ether = 1:8) to afford a yellow solid in an 87% yield (69 mg); mp 99–101 °C; ^1^H NMR (500 MHz, CDCl_3_) *δ* 9.10 (s, 1H), 7.78 (d, *J* = 7.0 Hz, 2H), 7.67 (s, 1H), 7.46–7.40 (m, 4H), 7.37 (d, *J* = 8.0 Hz, 1H), 7.33 (t, *J* = 8.0 Hz, 2H), 7.25–7.21 (m, 4H), 6.81 (d, *J* = 2.5 Hz, 1H); ^13^C NMR (125 MHz, CDCl_3_) *δ* 192.9, 139.6, 138.9, 133.9, 132.3, 131.8, 130.9, 130.7, 130.2, 130.1, 128.9, 128.8, 128.7, 128.4, 127.4, 123.6, 123.1, 122.4, 111.7; HRMS (ESI-TOF) *m/z*: [M + H]^+^ Calcd for C_23_H_17_BrNO 402.0488; found 402.0489.*(5-(2-Fluorophenyl)-2-phenyl-1H-pyrrol-3-yl)(phenyl)methanone* (**5m**). This compound was purified by column chromatography (ethyl acetate/petroleum ether = 1:8) to afford a yellow solid in a 79% yield (54 mg); mp 77–79 °C; ^1^H NMR (500 MHz, CDCl_3_) *δ* 9.39 (s, 1H), 7.81 (d, *J* = 7.0 Hz, 2H), 7.64 (t, *J* = 8.0 Hz, 1H), 7.48–7.44 (m, 3H), 7.34 (t, *J* = 8.0 Hz, 2H), 7.29–7.26 (m, 3H), 7.23–7.13 (m, 3H), 6.98 (d, *J* = 3.0 Hz, 1H); ^13^C NMR (125 MHz, CDCl_3_) *δ*192.8, 159.2 (d, *J_C-F_* = 124.1 Hz), 139.7, 138.3, 132.2, 132.0, 130.1, 128.8, 128.6, 128.57 (d, *J_C-F_* = 8.5 Hz), 128.4, 127.3 (d, *J_C-F_* = 4.0 Hz), 127.1, 125.3 (d, *J_C-F_* = 3.0 Hz), 121.7, 119.4, 119.3, 116.8 (d, *J_C-F_* = 23.8 Hz), 112.7; HRMS (ESI-TOF) *m/z*: [M + H]^+^ Calcd for C_23_H_17_FNO 342.1289; found 342.1281.*(4-Chlorophenyl)(2-(4-chlorophenyl)-5-phenyl-1H-pyrrol-3-yl) methanone* (**5n**). This compound was purified by column chromatography (ethyl acetate/petroleum ether = 1:8) to afford a yellow solid in a 75% yield (58 mg); mp 94–96 °C; ^1^H NMR (500 MHz, CDCl_3_) *δ* 8.75 (s, 1H), 7.77 (d, *J* = 9.0 Hz, 2H), 7.53 (d, *J* = 7.0 Hz, 2H), 7.46 (d, *J* = 8.5 Hz, 2H), 7.42 (t, *J* = 8.0 Hz, 2H), 7.34 (m, 4H), 6.80 (d, *J* = 2.5 Hz, 1H); ^13^C NMR (125 MHz, CDCl_3_) *δ* 191.2, 138.7, 138.0, 136.8, 134.8, 132.7, 131.4, 130.5, 130.0, 129.6, 129.2, 128.8, 127.9, 124.6, 122.3, 110.8; HRMS (ESI-TOF) *m/z*: [M + H]^+^ Calcd for C_23_H_16_Cl_2_NO 392.0603; found 392.0612.*(2,5-Diphenyl-1H-pyrrol-3-yl) (naphthalen-2-yl) methanone* (**5o**). This compound was purified by column chromatography (ethyl acetate/petroleum ether = 1:8) to afford a yellow solid in a 92% yield (68 mg); mp 107–109 °C; ^1^H NMR (500 MHz, CDCl_3_) *δ* 8.81 (s, 1H), 8.34 (s, 1H), 7.94 (d, *J* = 8.5 Hz, 1H), 7.83 (t, *J* = 8.0 Hz, 3H), 7.60–7.52 (m, 5H), 7.49 (d, *J* = 7.0 Hz, 1H), 7.42 (t, *J* = 8.0 Hz, 2H), 7.31–7.26 (m, 3H), 7.22 (d, *J* = 7.5 Hz, 1H), 6.92 (d, *J* = 2.5 Hz, 1H); ^13^C NMR (125 MHz, CDCl_3_) *δ* 192.7, 138.1, 137.0, 135.4, 132.7, 132.3, 132.2, 131.8, 131.7, 129.7, 129.5, 128.9, 128.8, 128.6, 128.2, 128.2, 128.1, 127.6, 126.8, 126.1, 124.5, 122.6, 111.0; HRMS (ESI-TOF) *m/z*: [M + H]^+^ Calcd for C_27_H_20_NO 374.1539; found 374.1537.*(2,5-Diphenyl-1H-pyrrol-3-yl) (thiophen-2-yl) methanone* (**5p**). This compound was purified by column chromatography (ethyl acetate/petroleum ether = 1:8) to afford a yellow solid in a 90% yield (59 mg); mp 86–88 °C; ^1^H NMR (500 MHz, CDCl_3_) *δ* 8.77 (s, 1H), 7.66 (d, *J* = 3.5 Hz, 1H), 7.60–7.54 (m, 5H), 7.42 (t, *J* = 8.0 Hz, 2H), 7.36 (t, *J* = 7.0 Hz, 2H), 7.33–7.28 (m, 2H), 7.04 (dd, *J* = 5.0, 4.0 Hz, 1H), 6.99 (d, *J* = 3.0 Hz, 1H); ^13^C NMR (125 MHz, CDCl_3_) *δ* 184.1, 145.9, 137.4, 134.1, 133.2, 132.4, 132.1, 131.8, 129.5, 129.0, 128.7, 128.6, 128.0, 127.6, 124.6, 122.3, 110.2; HRMS (ESI-TOF) *m/z*: [M + H]^+^ Calcd for C_21_H_16_NOS 330.0947; found 330.0955.*(2-Isopropyl-5-phenyl-1H-pyrrol-3-yl) (phenyl)methanone* (**5q**). This compound was purified by column chromatography (ethyl acetate/petroleum ether = 1:8) to afford a yellow solid in an 86% yield (50 mg); mp 85–87 °C; ^1^H NMR (500 MHz, CDCl_3_) *δ* 8.79 (s, 1H), 7.84 (d, *J* = 7.0 Hz, 2H), 7.53 (t, *J* = 7.0 Hz, 1H), 7.48–7.44 (m, 4H), 7.36 (t, *J* = 7.5 Hz, 2H), 7.23 (t, *J* = 7.5 Hz, 1H), 6.64 (d, *J* = 3.0 Hz, 1H), 3.87 (m, 1H), 1.37 (d, *J* = 7.0 Hz, 6H); ^13^C NMR (125 MHz, CDCl_3_) *δ* 192.8, 147.8, 141.1, 132.2, 131.6, 129.8, 129.5, 129.4, 128.5, 127.2, 124.3, 120.2, 110.1, 26.8, 22.4; HRMS (ESI-TOF) *m/z*: [M + H]^+^ Calcd for C_20_H_20_NO 290.1539; found 290.1530.*Phenyl(5-phenyl-2-propyl-1H-pyrrol-3-yl) methanone* (**5r**). This compound was purified by column chromatography (ethyl acetate/petroleum ether = 1:8) to afford a yellow solid in a 42% yield (24 mg); mp 74–76 °C; ^1^H NMR (500 MHz, CDCl_3_) *δ* 8.68 (s, 1H), 7.87–7.82 (m, 2H), 7.53 (t, *J* = 7.5 Hz, 1H), 7.49–7.44 (m, 4H), 7.39–7.34 (m, 2H), 7.23 (t, *J* = 7.5 Hz, 1H), 6.66 (d, *J* = 3.0 Hz, 1H), 3.02 (t, *J* = 7.5 Hz, 2H), 1.76 (m, 2H), 1.60 (s, 3H), 1.01 (t, *J* = 7.5 Hz, 3H); ^13^C NMR (125 MHz, CDCl_3_) *δ* 192.7, 142.4, 141.0, 132.1, 131.6, 130.0, 129.5, 129.4, 128.5, 127.1, 124.2, 121.2, 109.8, 30.1, 23.1, 14.4; HRMS (ESI-TOF) *m/z*: [M + H]^+^ Calcd for C_20_H_20_NO 290.1539; found 290.1542.*(2-Butyl-5-phenyl-1H-pyrrol-3-yl) (phenyl)methanone* (**5s**). This compound was purified by column chromatography (ethyl acetate/petroleum ether = 1:8) to afford a yellow solid in a 49% yield (30 mg); mp 74–76 °C; ^1^H NMR (500 MHz, CDCl_3_) *δ* 9.24 (s, 1H), 7.85 (d, *J* = 8.0 Hz, 2H), 7.55–7.45 (m, 5H), 7.34 (t, *J* = 7.0 Hz, 2H), 7.21 (t, *J* = 7.5 Hz, 1H), 6.67 (s, 1H), 3.01 (t, *J* = 7.5 Hz, 2H), 1.7–1.64 (m, 2H), 1.39–1.32 (m, 2H), 0.89 (t, *J* = 7.0 Hz, 3H); ^13^C NMR (101 MHz, CDCl_3_) *δ* 192.7, 142.7, 140.7, 131.8, 131.3, 129.8, 129.1, 129.0, 128.1, 126.6, 123.9, 120.6, 109.4, 31.7, 27.5, 22.6, 14.0, 13.9; HRMS (ESI-TOF) *m/z*: [M + H]^+^ Calcd for C_21_H_22_NO 304.1696; found 304.1702.*(1,2-Dimethyl-5-phenyl-1H-pyrrol-3-yl) (phenyl)methanone* (**7a**). This compound was purified by column chromatography (ethyl acetate/petroleum ether = 1:8) to afford a white solid in a 72% yield (40 mg); mp 96–98 °C; ^1^H NMR (500 MHz, CDCl_3_) *δ* 7.64 (d, *J* = 7.0 Hz, 2H), 7.26 (t, *J* = 7.5Hz, 1H), 7.14 (t, *J* = 7.5 Hz, 2H), 7.04 (q, *J* = 8.0 Hz, 4H), 6.98 (d, *J* = 6.5 Hz, 1H), 6.64 (s, 1H), 3.62 (s, 3H), 2.38 (s, 3H); ^13^C NMR (125 MHz, CDCl_3_) *δ* 194.4, 140.0, 135.8, 135.6, 131.9, 130.2, 128.8, 128.2, 128.0, 126.2, 125.9, 120.2, 120.1, 34.2, 11.7; HRMS (ESI-TOF) *m/z*: [M + H]^+^ Calcd for C_19_H_18_NO 276.1383; found 276.1388.*1-(2-Methyl-5-phenyl-1-(p-tolyl)-1H-pyrrol-3-yl) ethan-1-one* (**7b**). This compound was purified by column chromatography (ethyl acetate/petroleum ether = 1:8) to afford a yellow solid in an 85% yield (49 mg); mp 66–68 °C; ^1^H NMR (500 MHz, CDCl_3_) *δ* 7.38 (d, *J* = 4.0 Hz, 4H), 7.32–7.27 (m, 3H), 7.21 (d, *J* = 8.5 Hz, 2H), 6.64 (s, 1H), 2.42 (s, 3H), 2.39 (s, 3H), 2.07 (s, 3H); ^13^C NMR (125 MHz, CDCl_3_) *δ* 198.1, 138.5, 136.6, 136.5, 135.8, 130.3, 129.7, 128.7, 127.2, 126.6, 126.4, 122.8, 121.1, 31.5, 21.5, 13.3; HRMS (ESI-TOF) *m/z*: [M + H]^+^ Calcd for C_20_H_20_NO 290.1539; found 290.1538.*2-Methyl-4-phenyl-1H-imidazole* (**8**). This compound was purified by column chromatography (ethyl acetate/petroleum ether = 1:3) to afford a white solid in a 96% yield (30 mg); mp 57–59 °C; ^1^H NMR (500 MHz, CDCl_3_) *δ* 7.67 (d, *J* = 7.0 Hz, 2H), 7.37 (s, 1H), 7.30 (t, *J* = 7.5Hz, 2H), 7.13 (t, *J* = 7.5 Hz, 1H), 3.39 (brs, 1H), 2.29 (s, 3H); ^13^C NMR (125 MHz, DMSO-*d*_6_) *δ* 145.7, 138.2, 133.2, 129.1, 127.2, 125.1, 115.6, 14.2; HRMS (ESI-TOF) *m/z*: [M + H]^+^ Calcd for C_10_H_11_N_2_ 159.0917; found 159.091.*(E)-(2,5-Diphenyl-1H-pyrrol-3-yl) (phenyl)methanone oxime* (**9**). This compound was purified by column chromatography (ethyl acetate/petroleum ether = 1:8) to afford a white solid in a 94% yield (63 mg); mp 95–97 °C; ^1^H NMR (500 MHz, DMSO-*d*_6_) *δ* 11.44 (s, 1H), 11.19 (s, 1H), 7.79 (d, *J* = 7.5 Hz, 2H), 7.49 (d, *J* = 7.5 Hz, 4H), 7.37 (t, *J* = 8.0 Hz, 2H), 7.26–7.23 (m, 5H), 7.19 (t, *J* = 7.5 Hz, 1H), 7.12 (t, *J* = 7.5 Hz, 1H), 6.52 (d, *J* = 3.0 Hz, 1H); ^13^C NMR (125 MHz, DMSO-*d*_6_) *δ* 153.7, 137.7, 133.2, 132.8, 131.6, 129.6, 129.5, 129.1, 129.0, 128.8, 127.4, 127.2, 127.0, 126.5, 124.8, 115.1, 109.2; HRMS (ESI-TOF) *m/z*: [M + H]^+^ Calcd for C_23_H_19_N_2_O 339.1492; found 339.1496.*(1-Methyl-2,5-diphenyl-1H-pyrrol-3-yl) (phenyl)methanone* (**10**). This compound was purified by column chromatography (ethyl acetate/petroleum ether = 1:8) to afford a colorless oil in a 98% yield (66 mg); ^1^H NMR (500 MHz, CDCl_3_) *δ* 7.76 (d, *J* = 7.0 Hz, 2H), 7.50 (d, *J* = 8.0 Hz, 2H), 7.45 (t, *J* = 8.0 Hz, 2H), 7.40–7.35 (m, 5H), 7.34 (s, 1H), 7.32–7.27 (m, 3H), 6.67 (s, 1H), 3.49 (s, 3H); ^13^C NMR (125 MHz, CDCl_3_) *δ* 192.3, 140.8, 140.1, 135.7, 132.9, 132.3, 131.6, 131.2, 129.8, 129.4, 129.0, 128.6, 128.5, 128.1, 128.1, 122.3, 112.3, 34.3; HRMS (ESI-TOF) *m/z*: [M + H]^+^ Calcd for C_24_H_20_NO 338.1539; found 338.1544.

## 4. Conclusions

In conclusion, we have demonstrated that the Rh(II)-catalyzed substituent-controllable regioselective annulations provide a new synthetic strategy for trisubstituted imidazoles and pyrroles. The highlight of the current reaction is the substituent-dependent product selectivity. The imidazole skeleton was formed via N-H insertion to *α*-imino rhodium carbene, followed by intramolecular 1,4-conjugate addition when *α*-carbon atom of the amino group bore with methyl. Switching the methyl to phenyl group, the pyrrole framework was generated through N-H insertion and the intramolecular nucleophilic addition process. The large-scale reactions and transformations of the products further demonstrated the potential synthetic value of this strategy.

## Data Availability

The data presented in this study are available in this article.

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
