# Peer review of "Substituent-Controllable Cascade Regioselective Annulation of β-Enaminones with N-Sulfonyl Triazoles for Modular Access to Imidazoles and Pyrroles"

_molecules, 2023, doi:10.3390/molecules28114416_

Round 1

Reviewer 1 Report

The work of X. He et al. is devoted to the study of the reaction of triazoles with enaminoketones in the presence of a rhodium catalyst. It was shown that the nature of the substituents in the enaminoketones determines the pathway of transformations into imidazoles or pyrroles. This work is of interest for synthesists in the field of obtaining heterocyclic compounds, since functionalized heterocyclic systems are formed. 

Some typos and inaccuracies need to be corrected before the work can be recommended for publication.

Firstly, add CCDC numbers on schemes 2,3. On Scheme 4, correct the signature.

A question arose - why is the product 5q presented in Scheme 3, which has an alkyl substituent instead of an aryl substituent?

Scheme 4 - Caption to the mechanism includes isoquinolone, which is not on the Scheme, is this a mistake? In Scheme 4 the intermediates D, E are the same, what explanation do the authors offer to explain their further transformations into two different heterocycles?  It is not quite clear what is the driving force for the detachment of TsNH2 from G?

Author Response

Answers: We thank the reviewer for his/her time and effort in our manuscript.

1) We have checked the manuscript carefully, and modified the typos in the revised manuscript. 2) We have added the CCDC numbers on Scheme 2 and 3 for compounds 3a and 5a. We also corrected the signature on Scheme 4 in the revised manuscript.3) To further test the substrate scope of the β-enaminones, we introduced an alkyl substituent (such as a bulky isopropyl group replaces the aryl) to β-enaminone under the standard conditions. Satisfactory, the corresponding product 5q was isolated in 86% yield. 4) We apologize for our mistake of the caption of Scheme 6 (Scheme 4 is actually Scheme 6), and we have corrected the caption in the revised manuscript. 5) Based on the experimental results, we thought that it might be the steric hindrance of the substituents promoted the transformation of Intermediates D, E into two different heterocycles. Only the imidazole products were obtained when the amino carbon atom of β-enaminone bearing with the small steric methyl group. Similarly, the single pyrrole products were generated by replacing methyl with a bulky aryl or isopropyl group. Accordingly, introducing a moderate steric substituent, such as n-propyl or n-butyl, two desired products were isolated from the reaction system under the standard conditions. The corresponding explanation was also given in the description of Scheme 3-6 in the manuscript. 6) We think that intermediate G could convert to the aromatic product 5 under thermal conditions through the elimination process.

Reviewer 2 Report

The manuscript is well prepared. My only remark concerns the experimental part. I believe that the Experimental part should be placed in point 3. Materials and methods. Separating the experimental part as Appendix is unnecessary.

Please make the following corrections to the text:

Line 14: “of amino group” – change to “of the amino group”

Line 22: remove “of”

Lines 23-24: “framework” to “frameworks”

Line 32: “substituents,” - remove coma

Line 32: add coma after “compounds”

Line 36: “denitrogenative” change to “denitrogenation”

Line 43:  add “the” – “the regioselective”

Line 53: change “catalyst” to “catalysts”

Line 54: change “more” to “to be more”

Line 55: remove “at”

Line 58: “Further” change to “A further”

Lines 59-60: “Extension of reaction time did not find beneficial to …” reorganize this sentence: “Reaction time extension did not benefit …”

Line 61: remove “found to be”

Line 68: change ”set out to explore” to ”explored”

Line 71: “, and the results” change to “. The results” – make two sentences

Line 84: “viable” change to “a viable”

Line 92: “replace of methyl group” change to “replaced the methyl group”

Line 97: add coma after “e.g.”

Line 112: change “hypothesis” to “hypothesize”

Line 113: change “hinder” to “hindrance”

Line 124: change “an 75%” to “a 75%”

Line 135: it would be better to change “on the basis of” to “based on”

Line 141: please consider to change the end of the sentence: “with the regeneration of the rhodium(II) catalyst.” to “with the rhodium(II) catalyst regeneration.”

Line 142: change “taumerization” to “tautomerization”

Line 142: change “more” to “a more”

Line 144: add “in” after “results”

Line 158: change “amino” to “the amino”

Line 189: change “solvent to “a solvent”

Line 190: add “and” after coma

Line 194: change “compound” to “compounds”

Line 200: remove “atmosphere for”

Line 204: add coma after “Na2SO4”

Line 210: remove “atmosphere for”

Line 215: add coma after “Na2SO4”

Line 220: change “were” to “was” (because “A mixture”)

Line 220: change “reflux” to “a reflux”

Line 221: add coma after “added”

Line 222: change “in oil bath with stirring 12 hours” to “in an oil bath for 12 hours”

Line 224: add coma after “Na2SO4”

Line 229: add coma after “dropwise”

Author Response

Answers: We thank the reviewer for his/her time and effort in our manuscript.

We have moved the Experimental part to point 3 in the revised manuscript.

1) Line 14: We have changed “of amino group” to “of the amino group” in the revised manuscript.

2) Line 22: We have removed “of” in the revised manuscript.

3) Lines 23-24: We have corrected “framework” to “frameworks” in the revised manuscript.

4) Line 32: We have removed the comma after “substituents,” in the revised manuscript.

5) Line 32: We have added a comma after “compounds” in the revised manuscript.

6) Line 36: We have changed “denitrogenative” to “denitrogenation” in the revised manuscript.

7) Line 43: We have added “the” before “regioselective” in the revised manuscript.

8) Line 53: We have changed “catalyst” to “catalysts” in the revised manuscript.

9) Line 54: We have changed “more” to “to be more” in the revised manuscript.

10) Line 55: We have removed the “at” after “that” in the revised manuscript.

11) Line 58: We have changed “Further” to “A further” in the revised manuscript.

12) Lines 59-60: We have changed the text “Extension of reaction time did not find beneficial to …” to “Reaction time extension did not benefit …” in the revised manuscript.

13) Line 61: We have removed “found to be” in the revised manuscript.

14) Line 68: We have changed “set out to explore” to ”explored” in the revised manuscript.

15) Line 71: We have changed “, and the results” to “. The results” in the revised manuscript.

16) Line 84: We have changed “viable” to “a viable” in the revised manuscript.

17) Line 92: We have changed “replace of methyl group” to “replaced the methyl group” in the revised manuscript.

18) Line 97: We havd added a coma after “e.g.” in the revised manuscript.

19) Line 112: We have changed “hypothesis” to “hypothesize” in the revised manuscript.

20) Line 113: We have changed “hinder” to “hindrance” in the revised manuscript.

21) Line 124: We have changed “an 75%” to “a 75%” in the revised manuscript.

22) Line 135: We have changed “on the basis of” to “based on” in the revised manuscript.

23) Line 141: We have changed the text “with the regeneration of the rhodium(II) catalyst.” to “with the rhodium(II) catalyst regeneration.” in the revised manuscript.

24) Line 142: We have changed “taumerization” to “tautomerization” in the revised manuscript.

25) Line 142: We have changed “more” to “a more” in the revised manuscript.

26) Line 144: We have added “in” after “results” in the revised manuscript.

27) Line 158: We have changed “amino” to “the amino” in the revised manuscript.

28) Line 189: We have changed “solvent to “a solvent” in the revised manuscript.

29) Line 190: We have added “and” after coma in the revised manuscript..

30) Line 194: We have changed “compound” to “compounds” in the revised manuscript.

31) Line 200: We have removed “atmosphere for” in the revised manuscript.

32) Line 204: We have added a coma after “Na2SO4” in the revised manuscript.

33) Line 210: We have removed “atmosphere for” in the revised manuscript.

34) Line 215: We have added a coma after “Na2SO4” in the revised manuscript.

35) Line 220: We have changed “were” to “was” in the revised manuscript.

36) Line 220: We have changed “reflux” to “a reflux” in the revised manuscript.

37) Line 221: We have added a coma after “added” in the revised manuscript.

38) Line 222: We have changed “in oil bath with stirring 12 hours” to “in an oil bath for 12 hours” in the revised manuscript.

39) Line 224: We have added a coma after “Na2SO4” in the revised manuscript.

40) Line 229: We have added a coma after “dropwise” in the revised manuscript.

Reviewer 3 Report

This paper describes the synthesis of trisubstituted imidazoles and pyrroles through rhodium(II)-catalyzed regioselective annulation of N-sulfonyl-1,2,3-trizaoles with β-enaminones. Features such as mild conditions, good functional groups tolerance, gram-scale synthesis, and valuable transformations of the products qualified this unique protocol as an efficient tool for the synthesis of N-heterocycles. The mechanism of reaction proposed by the authors is a significant advantage. I strongly recommend the manuscript for publication. However, minor adjustments should be made before acceptance. They concern the description of 1H NMR and 13C NMR spectra. It is necessary to recalculate the signals in the spectra and check their correlation with the description. For example, in the description of the spectrum of compound 8, there should be signals for a total of 10 protons. Meanwhile, one proton is missing. Perhaps it is a proton bound to the N-1 nitrogen atom, but the fact that it is barely visible or not visible should be included. In addition, individual signals should be assigned not only the number of protons, but also structural fragments responsible for the appearance of a given signal. In descriptions of 13C NMR spectra, the number of signals does not agree with the number of carbon atoms in the molecule. This is probably due to the fact that coals with the same chemical environment give common signals. However, this should be indicated in the description of the spectrum. Check

English is clear but check all text for typos.

Author Response

Answers: We thank the reviewer for his/her time and effort in our manuscript.

We have checked the 1H NMR spectrum of compound 8, and found that the proton of N-H appeared as a very broad singlet around “3.39 (brs, 1H)”, and this information has been added to the revised manuscript. As the precise assignment of individual proton/carbon signal in NMR may require extensive NMR studies (such as HMQC, COSY, etc.), most current publications (including those published in the journal Molecules) have followed a practice not to make such assignments in reporting the NMR data, which we have also chosen to follow.
